

# Examining publication bias—a simulation-based evaluation of statistical tests on publication bias

Andreas Schneck

Department of Sociology, Ludwig-Maximilians-Universität München, Munich, Germany

## ABSTRACT

**Background**. Publication bias is a form of scientific misconduct. It threatens the validity of research results and the credibility of science. Although several tests on publication bias exist, no in-depth evaluations are available that examine which test performs best for different research settings.

**Methods**. Four tests on publication bias, Egger's test (FAT), p-uniform, the test of excess significance (TES), as well as the caliper test, were evaluated in a Monte Carlo simulation. Two different types of publication bias and its degree (0%, 50%, 100%) were simulated. The type of publication bias was defined either as *file-drawer*, meaning the repeated analysis of new datasets, or *p-hacking*, meaning the inclusion of covariates in order to obtain a significant result. In addition, the underlying effect ($\beta = 0$, 0.5, 1, 1.5), effect heterogeneity, the number of observations in the simulated primary studies ($N = 100$, 500), and the number of observations for the publication bias tests ($K = 100$, 1,000) were varied.

**Results**. All tests evaluated were able to identify publication bias both in the *file-drawer* and *p-hacking* condition. The false positive rates were, with the exception of the 15%- and 20%-caliper test, unbiased. The FAT had the largest statistical power in the *file-drawer* conditions, whereas under *p-hacking* the TES was, except under effect heterogeneity, slightly better. The CTs were, however, inferior to the other tests under effect homogeneity and had a decent statistical power only in conditions with 1,000 primary studies.

**Discussion**. The FAT is recommended as a test for publication bias in standard meta-analyses with no or only small effect heterogeneity. If two-sided publication bias is suspected as well as under *p-hacking* the TES is the first alternative to the FAT. The 5%-caliper test is recommended under conditions of effect heterogeneity and a large number of primary studies, which may be found if publication bias is examined in a discipline-wide setting when primary studies cover different research problems.

Corresponding author
Andreas Schneck,
andreas.schneck@lmu.de

## INTRODUCTION

All scientific disciplines try to uncover truth by systematically examining their surrounding environment (*Descartes, 2006*: 17). Natural scientists try to observe regularities in nature, whereas social scientists try to uncover patterns in the social behaviour of humans. The

success, as well as the reputation, of science rests on the accuracy and unbiasedness of scientific results. Publication bias, the publication of only positive results confirming the researcher's hypothesis (cf. *Dickersin & Min, 1993*: 135), threatens this validity. Under publication bias, only results showing either statistical significance and/or the desired direction of the effects are published. The published literature in this case is merely a selective (and too optimistic) part of all existing scientific knowledge. Furthermore, science is in the case of publication bias also inefficient as studies that add substantial knowledge to the literature, but contain null-findings remain unpublished.

The study at hand examines the performance of four methods to identify publication bias: Egger's Test/FAT (*Egger et al., 1997*; *Stanley & Doucouliagos, 2014*), p-uniform (PU; *Van Aert, Wicherts & Assen, 2016*; *Van Assen, Aert & Wicherts, 2015*), the test for excess significance (TES; *Ioannidis & Trikalinos, 2007*) and the caliper test (CT; *Gerber & Malhotra, 2008a*; *Gerber & Malhotra, 2008b*). In order to compare the performance of these tests, the false positive rate ($\alpha$-error, type I error) and the statistical power (true positive rate) were examined in a Monte Carlo simulation study. This makes it possible to assess the performance of the four tests under different conditions of publication bias (*file-drawer* vs. *p-hacking*), as well as study settings (underlying true effect, effect heterogeneity, number of observations in primary studies and in meta-analyses).

## The issue of publication bias

The false positive rate of a test (commonly called *p*-value) is the probability of the estimator rejecting $H_0$ despite this being true. The *p*-value is therefore the probability that the observed estimate is at least as extreme given there is no effect as assumed by $H_0$ (*Wasserstein & Lazar, 2016*). The larger the *p*-value the higher the risk of assuming an effect if none exists in the data. *p*-values below a certain threshold are called statistically significant, whereas values above the threshold are labelled as non-significant. In the empirical sciences the 5%-significance threshold is mostly used (*Cohen, 1994*; *Labovitz, 1972*; *Nuzzo, 2014*). The difference between 0.049 and 0.051 in the error probability is, however, marginal. Nevertheless, from the standpoint of the 5%-significance threshold the first would be a significant effect, whereas the latter would be a non-significant effect. In both of these two cases, on average around 1 in 20 null-hypotheses of no difference would be rejected, albeit true. If empirical researchers select their data/models until they find, just by chance, significant evidence that seems worth publishing, publication bias is on the rise, leading to inflated or even artificial effects.

*Rosenthal (1979)* constructs a worst case scenario in which only the 5% of false positive studies that are "significant" solely by pure chance are published. In this case, misinterpreted results shape the scientific discourse and finally result in (medical or political) interventions. Although Rosenthal's example is extreme, a multitude of evidence for publication bias exists in various disciplines and research fields (e.g., *Doucouliagos & Stanley, 2009*; *Jefferson et al., 2012*). *Godlee (2012)* therefore warns that scientific misconduct, under which publication bias is subsumed (*Chalmers, 1990*), may also physically harms patients.

In addition to the societal consequences, publication bias also has severe implications for the evolution of knowledge. Under publication bias no rejection of theories (*Popper, 1968*:

215), on which all scientific progress relies, occurs; this leads to a state of "undead theory" (*Ferguson & Heene, 2012*: 559) where all existing theories are confirmed irrespective of their truth.

## Motivation to commit publication bias

Because statistically significant results stress the originality of research findings (*Merton, 1957*), Both authors and scientific journals (cf. *Coursol & Wagner, 1986*; *Epstein, 1990*; *Epstein, 2004*; *Mahoney, 1977*) have large incentives to maximise their significant results to survive in a publish or perish research environment. Authors especially want to increase their publication chances, notably in top-tier journals where low acceptance rates of 5%–10% are quite common (for the top interdisciplinary journals *Nature, 2017*; *Science, 2017*; cf. for the political sciences *Yoder & Bramlett, 2011*: 266). Two distinct strategies to achieve significant results by means of publication bias practices can be pointed out. Firstly, non-significant findings can be suppressed (cf. the classical *file-drawer* effect described by *Rosenthal, 1979*) and significant results are then searched for in another dataset. Secondly, small bits in the data analysis can be changed (e.g., adding covariates, optional stopping, exclusion of outliers, etc.) until a significant result is obtained—this method is known as *p-hacking* (cf. "fishing" *Gelman, 2013*; or "researchers degree of freedom" *Simmons, Nelson & Simonsohn, 2011*: 1359). Whereas the *file-drawer* strategy can be utilised by authors as well as by editors and reviewers, *p-hacking* can only be committed by authors/researchers. Nonetheless, *p-hacking* strategies can be recommended by actors other than authors (e.g., editors, reviewers, etc.). Especially *p-hacking* is almost without any costs, as data analysis tools/packages become increasingly easy to apply (*Paldam, 2013*).

## Evidence on the prevalence of publication bias

So far, there are two strategies for identifying publication bias: the first traces studies through the publication process, the second asks authors, reviewers, or editors about their publication practices via surveys. In the first strategy, most of the analyses trace conference papers or ethics committee decisions if those results get published or remain in the file-drawer. Overall previous findings note, that studies with significant results have a substantially higher chance to get published (cf. *Callaham et al., 1998*; *Coursol & Wagner, 1986*; *Dickersin, 1990*; *Easterbrook et al., 1991*). *Ioannidis (1998)* in addition finds that significant studies have, beside their higher publication rate, also a substantially higher publication speed, meaning a shorter time between the completion of the study and the final publication. This results suggest that publication bias is a beneficial strategy in order to maximize academic merits.

The second approach asks directly about the publication practices of the involved actors. In a survey of psychologists that used a sensitive question technique up to 50% of the respondents claimed that they exercised publication bias (*John, Loewenstein & Prelec, 2012*: 525). *Franco, Malhotra & Simonovits (2014)* also note that most non-significant findings go to the file-drawer right after the analysis and are not even written up. Also, other forms of misbehaviour, like optional stopping (stopping data collection when significance is reached) or erroneous rounding of *p*-values to reach significant results, are alarmingly widespread

**Table 1 Publication bias tests in comparison.** compares the four evaluated publication bias tests in respect to four criteria, the measurement level, the sample used by the test, its underlying assumptions and its limitations.

| Test | Measurement level | Sample | Assumption | Limitation |
|---|---|---|---|---|
| FAT | Continuous $[-\infty, \infty]$ | All | Cov(es, se) = 0 | Only one-sided publication bias (PB) detectable |
| PU | Continuous $[0, 1]$ | $p < 0.05$, effects of same sign | Uniform or right skewed Skewness $\geq 0$ | Only one-sided PB detectable<br>Only on prespecified levels<br>Effect homogeneity (fixed-effect meta-analysis) |
| TES | Dichotomous $[0, 1]$ | All | E = O | Only on prespecified levels<br>Effect homogeneity (fixed-effect meta-analysis) |
| CT | Dichotomous $[0, 1]$ | Threshold $\pm$ caliper width | P(UC) = P(OC) | Only on prespecified levels |

(prevalence rate arround 22.5% *John, Loewenstein & Prelec, 2012*: 525). These results are in line with the survey of *Ulrich & Miller* (*2017*: 9), who report that researchers in the field of psychology prefer significant over non-significant results, and, furthermore, attribute more value to results with smaller *p*-values. These estimates may even be conservative because it is known from the survey literature that sensitive behaviours like scientific misconduct may be underreported (*Kreuter, Presser & Tourangeau, 2008*: 848). According to the presented research results *file-drawer* and *p-hacking* behaviour is therefore quite widespread.

## METHODS

### Publication bias tests in comparison

So far, the presented detection strategies ask either directly for publication preferences or examine the publication fate of conference papers. Both approaches have the weakness that they either rely on the potentially biased answers of the actors involved or require an immense effort to follow the publication process, while publication bias may have happened before the paper is submitted to a conference. Statistical tests on publication bias circumvent this problem by relying only on the published literature. In the paper at hand the regression-based FAT (*Egger et al., 1997*; *Stanley & Doucouliagos, 2014*), PU (*Van Aert, Wicherts & Assen, 2016*; *Van Assen, Aert & Wicherts, 2015*), an extended version of *p*-curve (*Simonsohn, Nelson & Simmons, 2014a*; *Simonsohn, Nelson & Simmons, 2014b*; *Simonsohn, Simmons & Nelson, 2015*), the TES (*Ioannidis & Trikalinos, 2007*), and the CT (*Gerber & Malhotra, 2008a*; *Gerber & Malhotra, 2008b*) were evaluated (see online Appendix for an in-depth discussion of the tests).[1]

In order to compare the different publication bias tests, four different criteria have to be established: the assumptions of the test, the measurement level, the sample used, the test method, and its according limitations (see Table 1).

The FAT tests basically the relationship between study's precision and its effect size with all available effect sizes from primary studies. If larger effects are observed for studies with low precision (and low *N*) publication bias is suspected. Nonetheless, alternative reasons may lead to this result: small studies examine specific high risk populations in which treatments may be more effective (*Sterne et al., 2011*); this effect heterogeneity may lead to the diagnosis of publication bias where none exists (*Schwarzer, Antes & Schumacher, 2002*).[2] The FAT has furthermore the disadvantage that only one-sided publication bias

[1]Because for Fail-save-N (*Rosenthal, 1979*) only rules of thumbs (instead of a formal statistical test) exist it was not included in the simulation at hand. Although it is still widely applied (*Banks, Kepes & McDaniel, 2012*: 183; *Ferguson & Brannick, 2012*: 4), this benchmark is not recommended in the *Cochrane Handbook*, a guideline for conducting meta-analyses (*Higgins & Green, 2008*: 321f.).

[2]For a similar result, see *Terrin et al. (2003)* for the related Trim and Fill technique (*Duval & Tweedie, 2000*).

[3]For simulations see: *Hayashino, Noguchi & Fukui (2005)*; *Kicinski (2014)*; *Macaskill, Walter & Irwig (2001)*; *Sterne, Gavaghan & Egger (2000)*.

[4]For simulations see: *Bürkner & Doebler (2014)*; *Kicinski (2014)*; *Moreno et al. (2009)*; *Renkewitz & Keiner (2016)*.

either in favour of a positive or negative significant effect can be tested. *Alinaghi & Reed* (*2016*: 10) show that if significant results of either sign are searched for, the FAT suffers from massively inflated false positive rates. In the Monte Carlo simulation at hand only the FAT is used because of its better statistical power as shown in prior simulations compared to the similar rank correlation test of *Begg & Mazumdar (1994)*[3] and the trim and fill technique (*Duval & Tweedie, 2000*).[4]

PU has the assumption that every left skewness in the distribution of *p*-values smaller than the significance threshold (e.g., $p < 0.05$) and conditioned on the underlying observed mean effect (*pp*-value) is caused by publication bias. This assumption is, however, grounded mainly on the fixed-effect estimate of the mean effect, which is very sensitive to effect heterogeneity. PU furthermore limits its test-value only on significant estimates in the direction where publication bias is suspected (*Van Aert, Wicherts & Assen, 2016*: 727). Therefore PU is, as the FAT, only able to identify one-sided publication bias.

The TES (*Ioannidis & Trikalinos, 2007*; also called ic-index see *Schimmack, 2012*) in contrast relies only on a dichotomous classifier, testing if the number of expected significant results and the empirically observed number of significant effects differ. Because the TES relies, as PU, on the fixed-effect estimate of the mean effect of all included studies it is sensible to effect heterogeneity. A large controversy in the literature is not about the TES itself, but on its application. Francis (*2012a*, *2012b*, *2012c*, *2012d*, *2012e*, *2013*) used the TES to identify singular articles in order to test if they suffer from publication bias. This may invalidate the assumption of independence (*Morey, 2013*: 181) as well as inflate the false positive rate in a similar manner than in primary research (cp. HARKing *Kerr, 1998*) if the TES is used in such an exploratory manner (*Simonsohn, 2013*: 175). Ioannidis, however, responds that if the TES is applied on prespecified research questions with a large and independent number of effect sizes, the TES is even a conservative test on publication bias (*Ioannidis, 2013*: 185).

The CT uses the most limited sample of the included tests that includes only estimates slightly over and under the chosen significance level in a distribution of *z*-values. In case of publication bias the assumption of a continuous distribution that results in an approximately even distribution in a narrow interval (caliper) is violated by an overrepresentation of just significant results. The broader the interval is set the more it may deviate from the assumed even distribution caused by the true underlying effect. This restrictive sample has the downside that the exclusion of most available values may drastically reduce the statistical power of the test.

In contrast to the FAT, the other tests are only able to test for publication bias on pre-specified levels (e.g., 0.05). Because the TES and the CT focus only on dichotomous classifiers (significant or not in the case of the TES, slightly over or under the threshold for the CT) also tests on two-sided publication bias are possible.

In previous simulation studies with a low number of included studies as well as observations PU was superior to the TES (*Van Assen, Aert & Wicherts, 2015*: 303) and the FAT (*Renkewitz & Keiner, 2016*). However, no evaluations exists based on a larger number of primary studies. In particular, the newer publication bias tests like PU, the TES, and the CT, are in need of an evaluation under different conditions. For the CT

**Table 2 Data generating process (DGP) of Monte Carlo simulation.** The 100 conditions of the Monte Carlo simulations are described. Two different aspects were varied: the underlying data and the publication bias behaviour of the actors. For the underlying data the true effect size, the number of observations ($N$) and the number of studies included in the meta-analysis ($K$) were varied. The behavioural component altered the proportions of authors who are willing to commit publication bias and its actual form as either *p-hacking* or *file-drawer*.

| Conditions | Values | Functional form | N (conditions) |
|---|---|---|---|
| **Data setup:** | | | |
| 1. True effects $\beta$: | $\beta = 0; 0.5; 1; 1.5;$ Het<br>$\varepsilon = NV(0,10)$<br>$\sigma_x = 2$ | $y = \beta x + \varepsilon$ | 5 |
| 2. Number of observations $N$: | $\mu_N = 100; 500$ | $|N(\mu_N)| N > 30$ | 2 |
| 3. Number of studies $K$: | $K = 100; 1,000$ | | 2 |
| **Behavioural setup:** | | | |
| 4. Publication bias (PB) | $PB = 0; 0.5; 1.0$ | $\beta > 0$ & $p < 0.05$ | $1 + 2*2 = 5$ |
| 4.1. *File-drawer* | Draw new sample size $N$ | *(max. 9 additional samples)* | |
| 4.2. *p-hacking* | Run new analyses with same dataset | $y = \beta x + \gamma_j z_j + \varepsilon$<br>$z = 0.5x + 0.5y + \varepsilon$<br>*(max. 3 z's = 7 combinations)* | |
| | | | $5*2*2*5 = 100$ |

also no studies exist regarding the best caliper width to use. Despite the existence of some simulation studies on publication bias tests, so far no direct comparison exists that evaluates the performance of all four publication bias tests, especially under effect heterogeneity.

## Simulation setup

In order to examine the performance of the four publication bias tests, a Monte Carlo simulation approach is used. For the simulation two different processes have to be distinguished: firstly, the data generation process (DGP), and, secondly, the meta-analytical estimation method (EM). The DGP provides the ground for the hypothetical data used by the simulated actors, as well as the results they report, whereas the EM applies the tests on publication bias reported in the previous section. The central advantage of using Monte Carlo simulations is that controlling the DGP allows to identify which simulated studies suffer from publication bias and which do not. Similar to the case in experiments, different conditions can be defined to ensure a controlled setting. The performance of the estimators can then be examined under the different conditions.

## Data setup of the primary studies and meta-analyses

The first step of the DGP defines different effect size conditions that underlie the analyses of the simulated actors (see Table 2). As a first condition, the underlying true effect was specified by a linear relationship with $\beta = 0, 0.5, 1.0, 1.5$. Analogous to a linear regression model, this means for $\beta = 0.5$ that an increase of one unit of the independent variable x increases the dependent variable $y$ by 0.5. The specified linear relationship between the dependent variable $y$ and the independent variable $x$ had a normally distributed regression error term of $\varepsilon = N(0,10)$, while the variation of the independent variable was defined as $\sigma_x = 2$ (for a similar setup see *Alinaghi & Reed, 2016*; *Paldam, 2015*). The regression coefficients can also be transformed in the Pearson correlation coefficient

[5]A total of 16.6% of the simulated studies were adequately powered with at least 80% power (*Cohen, 1988*: 56). The setting produced by the DGP also reflects the results of *Ioannidis, Stanley & Doucouliagos* (*2017*: 245), who report that only 10% of the studies in Economics are adequately powered.

yielding approximately $r = 0, 0.1, 0.2, 0.3$. This results are equivalent to low or medium effect sizes in terms of *Cohen (1992)* and cover about 75% of the empirical observed effects in psychology (*Bosco et al., 2015*: 436).[5] In addition to the homogenous conditions with a common effect size, a heterogeneous condition was added that assumes no fixed distribution of an underlying effect but a uniform mixture of all four effect sizes, as defined above, plus an additional effect of $\beta = 2.0$ ($r = 0.4$) in order to ensure enough variation.

As the FAT is based on study precision, which is mainly driven by the number of observations ($N$) of the primary studies, $N$ was computed as a second condition by an absolute normal distribution with a mean of 100 (small $N$) or 500 (large $N$) and a standard deviation of 150. In order to ensure an adequate statistical analysis for the primary studies, $N$ s equal to or smaller than 30 were excluded. This procedure resulted in a right skewed distribution with a mean $N$ of roughly 500 for the large $N$, and 165 for the small $N$ condition. The small $N$ condition reflects the observed number of observations in leading economics journals (mean: 152, own computations from the publicly available dataset of *Brodeur et al., 2016*) as well as of typical trials included in Cochrane reviews (mean: 118, *Mallett & Clarke, 2002*: 822). Because both studies refer mainly to an experimental literature, the large $N$ condition reflects the more common number of observations especially in ex-post-facto designs (e.g., survey studies).

The heterogeneity of effects in each of the meta-analyses was measured by $I^2$, the share of systematic variation in respect to the overall variation consisting of the systematic and random variation (*Higgins & Thompson, 2002*). In case of the small $N$ 68.62% and for a large $N$ 86.86% of the variation was systematic in the heterogeneous effect condition. In terms of *Higgins & Thompson* (*2002*: 1553), an $I^2$ larger than 50% has to be modelled explicitly in meta-analyses and cannot be ignored.

In addition to the number of observations in the primary studies ($N$) the number of primary studies that were included in the meta-analysis and form the basis of the publication bias tests ($K$) was varied in the third condition. A setting with 100 studies was used as a lower condition, whereas 1,000 studies were set as an upper condition. Although on average the number of trials in a meta-analysis is usually much lower than 100 studies (median 28 studies in the meta-meta-analysis by *Elia et al., 2016*: 5). One hundred studies were chosen because in this setting every publication bias test evaluated is at least partially applicable. In other research areas like Economics, where meta-regression models are more widely used to model effect heterogeneity, higher numbers of included trials estimates are also quite common (e.g., 1,474 effect estimates in *Doucouliagos & Stanley, 2009*).

### Behavioural setup of publication bias

Building on this data setup stage of the DGP the behavioural setup adds publication bias to the simulation in a fourth step (see Table 2). Publication bias was defined as the willingness to collect new data or run additional analyses if statistical significance failed ($p \geq 0.05$) or a negative effect occurred. In the simulation only one-sided publication bias was modelled because both the FAT and PU are not able to model two-sided publication bias that focuses only on significant results irrespective of its sign. It is important to note that only the intent to commit publication bias was varied in the simulation setup. The actual publication bias

depends on the data setup itself: how large is the true effect size ($\beta$) and the number of observations ($N$) in the primary studies? Or, in short: is there already a significant positive result which does not need a publication bias treatment?

Five different publication bias conditions have to be distinguished. Firstly, the condition without publication bias: in this ideal case all estimates ($\beta x$) are estimated by a bivariate ordinary least squares (OLS) model and afterwards published. Publishing in terms of the simulation model means that all estimates enter the final meta-analysis. Therefore, in the condition without publication bias either 100 or 1,000 regression results were estimated and enter the meta-analysis.

In the second and third conditions publication bias was present with a 50% probability. That means that 50% of the actors were willing to run additional analyses in order to obtain significant results. These conditions seem closest to the behavioural benchmark of the empirical studies presented.

If a non-significant result was obtained, actors operating under the second condition chose to collect new data in order to obtain significant results that can be published. This second condition therefore modelled publication bias under the *file-drawer* scenario, because the datasets not used remained unpublished. An actor tried to run analyses on the basis of up to nine additional datasets and only stopped earlier if a significant result with a positive sign was obtained. If none of the 10 datasets yielded a significant relationship with a positive sign, the estimate which was closest to the significance threshold has been published. This rule served two purposes: firstly, it seemed plausible that an actor who has tried that many analyses wants to get the results published in the end to compensate for the invested effort and to avoid sunk costs (*Thaler, 1980*). Secondly, from a technical point of view, this allowed to keep the number of observations in a meta-analysis $K$ constant across all simulation conditions.

In the third condition an actor did not try to achieve significant results by running the same bivariate analysis on different samples, but rather tried to run different model specifications on the same data by including control variables ($z_j$) to achieve statistical significance of the coefficient of interest ($\beta x$). The third condition therefore modelled publication bias as a form of *p-hacking*, because the existing dataset was optimised to receive a significant $p$-value. The actor was able to add three different control variables to the model. The control variables were defined as collider variables that are both an effect of $x$ as well as $y$, which biases the effect of interest (*Cole et al., 2009*; *Greenland, Pearl & Robins, 1999*). The effect of $x$ and $y$ on $z_j$ was, however, only small ($\gamma = 0.5$). The error term of the equation defining $z$ was normally distributed N(0,10). With three available control variables $z_j$ an actor had seven different combinations to improve the research results in order to obtain a significant effect of $x$ on $y$.

In contrast to the second and third conditions, where 50% of the actors had the intention to commit publication bias, in the fourth and fifth conditions all actors had the intention to engage in publication bias practices, once again either through *file-drawer* (fourth condition) or *p-hacking* behaviour (fifth condition). Part from the higher degree of intention to engage in publication bias practices the settings remained the same. Although the two conditions where all actors had the intent to engage in publication bias are far
[6]In order to specify the number of replications that are necessary to achieve a sufficient statistical power of at least 80% (*Cohen, 1988*: 56), a power analysis was conducted for the statistical power as well as the false positive rate estimates. For the false positive rate, a small deviation of 1 percentage point from the set 5%-false positive rate has to be correctly identified with at least an 80% chance. To achieve this goal, every condition without publication bias had to be supported with 3,729 runs. As deviations in power are, though important but not as essential as the false positive rate (*Cohen, 1988*: 56) a difference of 3 percentage points is set as acceptable. In order to identify a 3 percentage point deviation from the target power of 80% each of the 80 conditions with existing publication bias needed 1,545 runs. In total, 198,080 runs were necessary, resulting in nearly 109 million primary studies that in the case of publication bias contained up to 10 different regression models.

too pessimistic, they allow to evaluate the performance of the tests in the most extreme publication bias environment. Tests that are not able to detect publication bias even under such extreme conditions are of low utility to the research community.

The resulting design matrix had 100 different combinations resulting from 20 data setup conditions multiplied by the five publication bias conditions. In order to obtain reliable estimates similarly to an experiment (*Carsey & Harden, 2013*: 4f.), every single cell of the design matrix had to be replicated multiple times.[6]

The aim of the simulation study at hand was to compare the performance of the four tests in respect of: (A) their capability to detect publication bias if present (true positive, statistical power), as well as (B) consistent false positive classification ($\alpha$-error). Because the conditions with and without publication bias are known in a simulation study, the power of the tests and the false positive rate is computable (*Mooney, 1997*: 77–79). In a first step, a dummy variable ($s$) was constructed, with the value 1 for a significant test result below the significance threshold (5% significance level; $s = 1$ if $p < 0.05$). The statistical power, was defined as the proportion of significant results $s$ in respect to all *runs* ($r$) with publication bias ($\sum_{i=1}^{r} s_i / r$ *if* $PB > 0$). The false positive rate was computed equivalently but in conditions without publication bias ($\sum_{i=1}^{r} s_i / r$ *if* $PB = 0$).

## RESULTS

### Prevalence of publication bias

Because publication bias in the experimental setup was implemented as the intent to commit publication bias, three variables are useful to address the actual publication bias and its impact on the overall bias. Firstly, the share of actual studies per meta-analysis that suffer from publication bias (if $p < 0.05$ or negative result are obtained as a first result), secondly the share of studies that achieve their goal of a significant positive result by publication bias, and thirdly the impact of publication bias on the $p$-value of a fixed-effect meta-analysis (deflation factor of the $p$-value). Because the heterogeneous effect condition of the simulation does not allow an absolute bias measure the $p$-value deflation factor was used for all conditions.

In a first step the focus is on how the opportunity structures of the simulation conditions shape the committed publication bias. In the first two columns of Table 3 the actual committed publication bias is shown dependent on study characteristics like the mean number of observations in the primary studies ($N$) and the underlying effect ($\beta$), including effect heterogeneity. As expected, around 50% respectively 100% of the studies committed publication bias in case of an underlying null-effect because only 2.5% of the results had the right positive sign and were significant just by chance. In case of an underlying effect the share of committed publication bias decreased because an already significant finding made a publication bias treatment unnecessary. For $\beta = 0.5$ in the 50% publication bias condition only 35%; for $\beta = 1$, 15% for $\beta = 1.5$ only 9%; and in the heterogeneous condition 22% of the studies employed publication bias practices. The 100% publication bias condition approximately doubled the prevalence rates of the 50% condition as expected.

**Table 3  Risk factors for publication and its impact on bias in the simulated data (OLS regression).** The first two columns in Table 3 show that actual committed publication bias behaviour depended largely on the opportunity structure of the underlying data. Despite the defined 50% or 100% willingness of the authors to commit publication bias, only those actors who face insignificant effects (caused by small effects and sample sizes) engaged in publication bias practices. The success of publication bias in terms of significant results is shown in column three, dependent on the opportunity structure and form of publication bias. Conditions under p-hacking were slightly less effective in obtaining significant results than conditions under file-drawer publication bias. Column four shows the deflating impact of publication bias on meta-analytic p-values. For an average publication bias this p-values halved or even quartered.

| | Publication bias committed (50% intention) | Publication bias committed (100% intention) | Publication bias successful (in relation to committed) | Deflation of $p$-value |
|---|---|---|---|---|
| $N = 500$ | $-0.105^{***}$ | $-0.211^{***}$ | $0.179^{***}$ | |
| (ref. $N = 100$) | (0.000) | (0.001) | (0.001) | |
| $\beta = 0.5$ | $-0.196^{***}$ | $-0.391^{***}$ | $0.471^{***}$ | |
| (ref. $\beta = 0$) | (0.001) | (0.001) | (0.001) | |
| $\beta = 1$ | $-0.389^{***}$ | $-0.777^{***}$ | $0.513^{***}$ | |
| | (0.001) | (0.001) | (0.001) | |
| $\beta = 1.5$ | $-0.451^{***}$ | $-0.899^{***}$ | $0.503^{***}$ | |
| | (0.001) | (0.001) | (0.002) | |
| $\beta =$ heterogeneous | $-0.319^{***}$ | $-0.636^{***}$ | $0.358^{***}$ | |
| | (0.001) | (0.001) | (0.001) | |
| p-hacking | | | $-0.100^{***}$ | $0.197^{***}$ |
| (ref. file-drawer) | | | (0.001) | (0.002) |
| Committed PB [ +10ppts] | | | | $-0.018^{***}$ |
| (ref. mean $= 32.5\%$) | | | | (0.001) |
| Successful PB [+10ppts] | | | | $-0.077^{***}$ |
| (ref. mean $= 18.8\%$) | | | | (0.001) |
| Constant | $0.541^{***}$ | $1.080^{***}$ | $0.313^{***}$ | $0.225^{***}$ |
| | (0.000) | (0.001) | (0.001) | (0.002) |
| Observations | 61,760 | 61,760 | 115,843 | 123,520 |
| $R^2$ | 0.939 | 0.958 | 0.648 | 0.168 |

**Notes.**
Standard errors in parentheses.
$^{*}p < 0.05$.
$^{**}p < 0.01$.
$^{***}p < 0.001$.

Besides the necessity of publication bias to achieve significant results also the success probability in respect to committed publication bias depended on the conditions of the primary studies as shown by the third column of Table 3. For small studies ($N = 100$) with an underlying null-effect ($\beta = 0$) the success-probability in respect to the committed publication bias was about 31.3%. Publication bias got more effective if larger studies ($N = 500$) provide the primary study with more statistical power. The success probability of publication bias rose dramatically around 50 percentage points if a specific underlying empirical effect existed. Also, in case of effect heterogeneity, the success probability increased about 35.8 percentage points. Slight differences could be observed in the effectivity

of the publication bias mechanism, as *p-hacking* was with 10 percentage points less effective than the *file-drawer* condition to achieve significant results.

As publication bias deflates *p*-values and therefore biases meta-analytical effect estimates the impact of the actual observed publication bias (the share of committed and successful publication bias) on the meta-analytical *p*-value is presented. The fourth column of Table 3 shows that with an average proportion of publication bias committed (32.6%) as well successfully implemented (18.8%) in a meta-analysis with 100 studies ($K = 100$) the *p*-value of the meta-analysis more than quartered. This is further aggravated if the share of committed as well as successful publication bias rose by 10 percentage points. The actual impact of successful publication bias deflated the *p*-values by 7.7 percentage points and was more pronounced than the deflation caused by non-successfully committed publication bias (deflation by 1.7 percentage points). The deflation also was less severe if *p-hacking* procedures, as implemented in the simulation at hand, were used. Nonetheless, the meta-analytical *p*-value in case of *p-hacking* is still less than half the size (42.2%) of the unbiased estimate.

## False positive rate of publication bias tests

For the evaluated tests on publication bias consistent false positive rates are most important. In the simulation none of the tests should exceed the prespecified 5% error probability in any condition. The false positive rate of the test was fixed in the simulation setting to 0.05, so all false positive rates should be equal to, or even smaller than, 0.05. Positive deviations from 0.05 point to inflated false positive rates, which lead to more false conclusions than expected.

Table 4 shows the false positive rate in dependence of the number of studies included ($K = 100, 1,000$) and effect heterogeneity measured by $I^2$. In the constant condition of a meta-analysis with $K = 100$ and no effect heterogeneity none of the tests had larger false positive rates than the expected 0.05. In particular, the TES, the 3% and 5% CTs were very conservative. A larger meta-analytical sample increased the false positive rates for the TES and the CTs. The broadest 15% CT missed the expected significance threshold of 5%, with 7.8% clearly. The false positive rates for PU in contrast were slightly lower. Increasing effect heterogeneity resulted in more conservative false positive rates for PU, the 15% CT, and to a smaller extent also for the FAT, the TES and the 10% CT. The narrower 3% and 5% CTs were unaffected by effect heterogeneity.

The overall influence of the varied conditions on the false positive rate was small, as can be seen by the small share of explained variance ($R^2 < 1.7\%$). Looking at the false positive rates by each condition (Table A1 in the online Appendix) only the 10%- and 15%-caliper showed increased false positive rates because the underlying true effect rather than publication bias elicited an overrepresentation of just significant values. Note however, that the 3% and 5% CTs showed no increased false positive rates.

## Statistical power of publication bias tests

The following regression model (Table 5) addresses the statistical power conditional on the type of publication bias and its occurrence (committed as well as successful publication

**Table 4  Conditional false positive rates of the publication bias tests (OLS regression).** Table 4 displays the false positive rates of the publication bias tests conditional on the number of studies included in the meta-analysis ($K$) as well as the between study heterogeneity ($I^2$). The FAT had the most consistent false positive rate. The 15% CT missed the 5%-level clearly while the 10% CT showed a large variability and gets close to it. The 10% and 15% CT are therefore problematic because they may suffer from inflated false positive rates.

| | PU | FAT | TES | 3% CT | 5% CT | 10% CT | 15% CT |
|---|---|---|---|---|---|---|---|
| $K = 1{,}000$ | −0.005[***] | 0.000 | 0.006[***] | 0.026[***] | 0.023[***] | 0.026[***] | 0.050[***] |
| (ref. $K = 100$) | (0.001) | (0.002) | (0.001) | (0.001) | (0.001) | (0.001) | (0.002) |
| $I^2$ [ +10 percentage points] | −0.003[***] | −0.001[***] | −0.001[***] | 0.000 | 0.000 | −0.001[***] | −0.003[***] |
| | (0.000) | (0.000) | (0.000) | (0.000) | (0.000) | (0.000) | (0.000) |
| Constant[a] | 0.023[***] | 0.049 | 0.010[***] | 0.003[**] | 0.008[***] | 0.019[***] | 0.028[***] |
| | (0.001) | (0.001) | (0.001) | (0.001) | (0.001) | (0.001) | (0.001) |
| Observations | 73,960 | 74,560 | 74,560 | 62,644 | 66,546 | 69,718 | 70,936 |
| $R^2$ | 0.005 | 0.000 | 0.002 | 0.010 | 0.007 | 0.006 | 0.017 |

**Notes.**
[a]Test H0: constant = 0.05.
Standard errors in parentheses.
[*]$p < 0.05$.
[**]$p < 0.01$.
[***]$p < 0.001$.

bias). Starting from the baseline condition of a meta-analysis with $K = 100$, a mean share of publication bias committed (32.6%), as well as successfully applied (18.8%) via a *file-drawer* procedure and no effect heterogeneity, the FAT had a superior power of 56.9%, followed by the TES (51.5%) and the PU (48.3%). The CTs performed worst and yielded only a power of 0.0%–38.6%.

The underperformance of the CTs is largely explained by the small number of studies in the meta-analyses. With $K = 100$ hardly any study falls within the small caliper around the significance threshold. This limitation on just significant or non-significant effects also led to missing values, because without observations in the caliper no CT could be performed. The underperformance of the CT changed if 1,000 studies were included, which improved the estimated power substantially, by 30.7–57.3 percentage points, while smaller calipers profited most. The FAT, the TES, and the PU, profited moderately from an increased number of studies, by 24.4, 23.8 and 16.5 percentage points, respectively. When focussing on the influence of heterogeneity in the meta-analyses the PU and the TES showed a drastic drop in power, by 6.5 and 6.4 percentage points, if the heterogeneity measured by $I^2$ rose by 10 percentage points. This decrease in power shows that neither PU nor TES were able to cope with heterogeneity. In contrast, the FAT and the CTs actually showed a slight increased statistical power. Varying the publication bias procedure from *file-drawer* to *p-hacking,* which is less related to the standard error of the effect estimates, increased the power of PU, TES, and the CTs. The CTs profited most, increasing the statistical power by around 18 percentage points. The TES and PU showed a smaller increase of power, by 7.5 and 4.8 percentage points. The FAT, in contrast lost about 11 percentage points of its power under *p-hacking* compared to the *file-drawer* condition. Although the differences in power are dependent especially on the operationalization of the *p-hacking* condition in the simulation, this result points on a weakness of the FAT under non *file-drawer* conditions

**Table 5 Conditional statistical power of the publication bias tests (OLS regression).** shows the statistical power of the publication bias tests conditional on the number of studies included in the meta-analysis ($K$) and the between study heterogeneity ($I^2$). In contrast to Table 4, also the share of committed as well as successful publication bias and its form as either file-drawer or *p-hacking* was controlled. Overall the FAT had the largest power but was not able to detect *p-hacking* as good as the TES. The CTs were underpowered if a low number of studies was included in a meta-analysis but performed well in studies with large Ks. Both, PU and the TES, were not able to detect publication bias under effect heterogeneity.

| | PU | FAT | TES | 3% CT | 5% CT | 10% CT | 15% CT |
|---|---|---|---|---|---|---|---|
| $K = 1,000$ | 0.165*** | 0.244*** | 0.238*** | 0.573*** | 0.513*** | 0.382*** | 0.307*** |
| (ref. $K = 100$) | (0.002) | (0.002) | (0.002) | (0.002) | (0.002) | (0.002) | (0.002) |
| $I^2$ [+10 percentage points] | −0.065*** | 0.001* | −0.064*** | 0.006*** | 0.005*** | 0.008*** | 0.010*** |
| | (0.000) | (0.000) | (0.000) | (0.000) | (0.000) | (0.000) | (0.000) |
| *p-hacking* | 0.048*** | −0.110*** | 0.075*** | 0.179*** | 0.187*** | 0.186*** | 0.177*** |
| (ref. file-drawer) | (0.002) | (0.002) | (0.002) | (0.002) | (0.002) | (0.002) | (0.002) |
| Comitted PB [+10ppts] | 0.051*** | 0.030*** | −0.065*** | −0.035*** | −0.053*** | −0.073*** | −0.084*** |
| (ref. mean = 32.6%) | (0.000 | (0.001) | (0.001) | (0.001) | (0.001) | (0.001) | (0.001) |
| Successful PB [+10ppts] | 0.103*** | 0.099*** | 0.221*** | 0.162*** | 0.193*** | 0.224*** | 0.234*** |
| (ref. mean = 18.8%) | (0.001 | (0.001) | (0.001) | (0.001) | (0.001) | (0.001) | (0.001) |
| Constant[a] | 0.483*** | 0.569*** | 0.515*** | −0.002 | 0.125*** | 0.300*** | 0.386*** |
| | (0.002) | (0.002) | (0.002) | (0.002) | (0.002) | (0.002) | (0.002) |
| Observations | 123,520 | 123,520 | 123,520 | 107,736 | 111,315 | 115,243 | 117,207 |
| $R^2$ | 0.572 | 0.306 | 0.473 | 0.497 | 0.483 | 0.457 | 0.446 |

**Notes.**
[a] Test H0: constant = 0.8.
Standard errors in parentheses.
*$p < 0.05$.
**$p < 0.01$.
***$p < 0.001$.

[7] Significant or not (TES) over- or under-caliper (CTs).

that are less related to the standard error of the estimate but are still detectable in the distribution of $z$- or $p$-values.

The structural difference between tests based on a continuous effect distribution (FAT, PU) and tests that focus only on a dichotomous classification (TES, CTs)[7] becomes evident looking at the effect of the proportion of studies that underwent a publication bias treatment in the simulation and the proportion of studies that had a successful outcome after publication bias. Increasing the share of studies under publication bias lifted the power by 3.0 (FAT) and 5.1 (PU) percentage points. A 10 percentage point increase in studies successfully applying publication bias increases the power by 9.9 (FAT) and 10.3 percentage points (PU). The TES and the CTs, however, were only able to detect successful publication bias. An increase only in studies committing publication bias (whether successful or not) in contrast reduced the statistical power. Both tests were therefore not able to detect all possible outcomes of publication bias. This is especially problematic as non-successful publication bias may also inflate the overall estimated effect in meta-analyses. All effects presented are statistically significant ($p < 0.05$).

In contrast to the influence of the varied conditions on the false positive rate, the influence on statistical power was substantial, varying from 30.6% in the case of the FAT to 57.2% for the PU. This finding underlines the fact that all publication bias tests have their strengths and weaknesses in specific conditions.

## DISCUSSION & CONCLUSION

In the simulation at hand, the performance of four different tests (PU, FAT, TES, CTs) were evaluated in a Monte Carlo simulation. Different conditions were varied: the underlying true effect size, including effect heterogeneity, the number of observations in the primary studies, the number of studies in the meta-analyses, the degree of publication bias and its form as either *file-drawer* or *p-hacking*.

### Limitations

In order to compare the tests in a realistic setting that is nonetheless at least from the assumptions of all four tests applicable, four central limitations have to be pointed out:

Firstly the simulation and its according publication bias procedures rest on the assumptions that all the correlation between the study's precision and its effect size is caused by publication bias. In case that studies with larger effects are, for example after a pre-study power analysis (*Lau et al., 2006*) conducted with a lower number of observation especially the FAT may yield increased false positive rates (*Schwarzer, Antes & Schumacher, 2002*).

Secondly the number of observations included in the meta-analyses either set to $K = 100$ or 1,000 is large compared to the average meta-analysis (*Elia et al., 2016*). The results however showed that even in such large meta-analyses and especially in the more realistic condition in which 50% of the actors are willing to commit publication bias, the tests hardly yielded an adequate statistical power under most conditions. Increasing the number of included studies is therefore important to assure an adequately powered test on publication bias.

Thirdly the analysis focused only on one specific form of *p-hacking* that could occur in both small ($N = 100$) or large studies ($N = 500$). Especially for studies where $N$ is small, other strategies like optional stopping may also be applied. Further research on publication bias should therefore focus on the different impact of other *p-hacking* practices.

As a fourth limitation only one-sided publication bias against insignificant or negative results was simulated. By assumption especially PU limits only on the negative or positive signed studies that were supposed to be affected by publication bias. Also the FAT is not able to detect two-sided publication bias because the funnel in this situation may still be perfectly symmetric. The suggestions for applications if two-sided publication bias is suspected are therefore limited to the TES and the CTs only.

### CONCLUSION

The following five conclusions can be derived from the results: Firstly, for homogenous research settings and with publication bias favouring only effects in one direction (one-sided publication bias) the FAT is recommended due to its most consistent false positive rate as well as its superior statistical power. Secondly, if there are concerns whether there are any correlations between the precision of the study and its effect size for other reasons than publication bias (see first limitation) and if *p-hacking* is suspected, the TES should be preferred to the FAT under effect homogeneity. As the 5% CT offers more relaxed assumptions it is therefore the first alternative for the FAT under effect heterogeneity if a large number of studies is included in the meta-analysis.

Despite the analysis focussed only on one-sided publication bias, also two-sided publication bias, favouring significant results with either sign may also be present. As PU and the FAT are not able to identify one-sided publication bias only the TES and the CTs remain for two-sided publication bias. Therefore, fourthly, the TES is recommended under effect homogeneity because of its larger statistical power compared to the CTs. Fifthly, in the case of heterogeneous effect sizes and a sufficient number of observations in the meta-analysis the 5% CT provides the best trade-off between a conservative false positive rate and a decent statistical power.

The 5% CT is therefore best used to identify publication bias in an effect heterogeneous discipline-wide setting which relies per definition on completely different underlying effects but offers enough studies to compensate for the low statistical power. Because the wider 10% and 15% CTs yield inflated false positive rates, at least in some conditions, they are not recommended to identify publication bias.

Identifying publication bias in substantial meta-analyses as well as focussing on publication as a general problem within the scientific domain is necessary in order to establish and retain trust in scientific results. Further research, however, should not only focus on the diagnosis of publication bias just stating a problem that is well known (*Morey, 2013*). Beyond the nonetheless important diagnosis of the scientific "disease" a further examination of the risk factors, either on the side of the involved actors or with regard to the incentive structure within the discipline (see for example *Auspurg & Hinz, 2011*) seem essential. This includes also the evaluation of possible interventions (e.g., an open data policy). Research on publication bias is inevitable to maintain trust in scientific results and avoid wasted research funds that also limit the efficiency of science as a whole.

Beside the diagnosis of publication bias and its risk factors, also estimators of the unbiased effect, that are beyond the scope of this paper, like the effect estimates provided by PU and the PET (for example the PET/PEESE procedure of *Stanley & Doucouliagos, 2014*) should be evaluated comparatively. This is at most important for meta-analyses with a heterogeneous effect that try to uncover the underlying true effect rather than test for publication bias alone.

## ACKNOWLEDGEMENTS

I thank Katrin Auspurg, the two reviewers, Joseph Hilgard, Bob Reed and the responsible editor Robert Winkler for their valuable comments on this article.

### Funding
The authors received no funding for this work.

### Competing Interests
The authors declare there are no competing interests.

## Author Contributions

- Andreas Schneck conceived and designed the experiments, performed the experiments, analyzed the data, contributed reagents/materials/analysis tools, wrote the paper, prepared figures and/or tables, reviewed drafts of the paper.

## Data Availability

Further information on the analyses is contained in the readme.txt contained in the Supplemental zip File.

## Supplemental Information

Supplemental information for this article can be found online at http://dx.doi.org/10.7717/peerj.4115#supplemental-information.

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
