# Peer review of "Examining publication bias—a simulation-based evaluation of statistical tests on publication bias"

_PeerJ, doi:10.7717/peerj.4115_

## Round 0.1 · original submission · Major Revisions

· Academic Editor

Major Revisions

There were some concerns about the conditions/ assumptions used in your simulations/ analyses. I would be critical to resolve these concerns. Further, the manuscript would benefit from a more concise / clear presentation of the findings.

·

Basic reporting

Please see report.

Experimental design

Please see report.

Validity of the findings

Please see report.

Additional comments

General Comments:

This paper uses Monte Carlo simulations to test four different methods to identify publication bias: (i) Egger’s Test/Funnel Asymmetry Test (FAT), (ii) the p-uniform test (PU), (iii) the test for excess significance (TES), and (iv) the caliper test (CT – where 4 different versions of the CT are compared). The Monte Carlo experiments vary over a number of dimensions: (i) existence and type of publication bias (“p-hacking”; “file drawer”), (ii) number of observations in primary studies (N); (iii) number of studies included in simulated meta-analyses (K); and (iv) population effect size (both fixed at different values, and heterogeneous). Tests were compared on two dimensions, Type I error rates and power. As the author notes, there are a wide variety of results, making it difficult to come up with a single conclusion. But given the interest in identifying “best practice,” the author concludes: “The FAT is recommended as a test for publication bias in standard meta-analyses with no or only small effect heterogeneity. If no clear direction of publication bias is suspected the TES is the first alternative to the FAT. The 5%-caliper tests is recommended under conditions of effect heterogeneity, which may be found if publication bias is examined in a discipline-wide setting when primary studies cover different research problems” (cf. “Discussion”). Overall, the paper is clear and well-constructed. It makes a valuable contribution to an important topic about which relatively little is known. In this reviewer’s opinion, there is insufficient evidence to support the recommendations. However, the paper begins to fill in a blank sheet and hopefully moves the discipline closer to a better understanding of this issue. Specific comments follow.

Specific Comments:

1. The paper attempts to construct realistic situations by which to support recommendations for meta-analysis researchers. In the opinion of this reviewer, the paper is too narrow to do that. Most of the emphasis in the paper is based on fixed effects models. There is only one variation of a heterogeneous effect, the case identified as “Het.” As this latter case surely most matches realistic situations, there is an insufficient database of realistic situations on which to support recommendations. This reviewer is not recommending that the author change her Monte Carlo experiments. The fixed effect cases are of interest as a benchmark. Instead, this reviewer recommends that the conclusion be rewritten to downplay recommendations and focus more on what we can learn from this narrow set of experiments.

2. One important issue that is overlooked in this and related studies is the relationship between effect size and publication bias. The author seems to think that the 50% and 100% publication bias regimes are holding constant the degree of publication selection bias. They are not. As effect sizes increase, the degree of publication selection bias changes. This is evident in the power results in Tables 7-10. Again, this reviewer does not suggest that the author change her Monte Carlo experiments. However, it would be useful if the “percent positive and significant studies” was reported as a separate column for each of the “N/K/Effect size” experiments.

3. Relatively little analysis is done of the results. This author needs to be more thoughtful about explaining why results differ across N/K/Effect Size (and “percent positive and significant studies”/incidence of publication selection bias). In this reviewer’s opinion, the major contribution of this study relies on the insights that can be learned about how these parameters affect the Type I error rates and power results.

4. Related to the previous point, more effort needs to be paid to how the publication selection process is likely to affect the results across different parameter settings. For example, in the Reed (2015) and Alinaghi and Reed (2016) papers cited by the author, it is noted how effect size is related to the incidence of publication selection bias, and how this generates different kinds of estimation bias depending on the type of publication selection bias (bias against insignificance, bias against negative signs). The author has chosen to model a publication selection process that simultaneously targets both the sign of the estimate and its statistical significance. This is common across simulation studies (such as any of the many Stanley and Doucouliagos studies), so this reviewer has no problem with this approach. However, it does complicate the relationship between the incidence of publication selection and the degree of estimation bias, and hence Type I error rates and power and thus it will be that much more challenging, and important, that the author explain this relationship.

5. The perils of implementing these tests with heterogeneous effects should be emphasized. In that context, the following references should be discussed and cited:

-- Lau, J., Ioannidis J.P.A., Terrin, N., Schmid, C.H., and Olkin, I. (2006). The case of the misleading funnel plot. British Medical Journal 333: 597-600.

-- Sterne, J.A.C., Sutton, A.J., Ioannidis, J.P.A., Terrin, N., Jones, D.R., Lau, J., Carpenter, J., Rucker, G., Harbord, R.M., Schmid, C.H., Tetzlaff, J., Deeks, J., Peters, J., Macaskill, P., Schwarzer, G., Duval, S., Altman, D.G., Moher, D., and Higgins, J.P. (2011). Recommendations for examining and interpreting funnel plot asymmetry in meta-analyses of randomised controlled trials. British Medical Journal, 343: d4002.

-- Terrin, N., Schmid, C.H., Lau, J. and Olkin, I., (2003). Adjusting for publication bias in the presence of heterogeneity. Statistics in Medicine, 22: 2113-2126.

·

Basic reporting

I was able to load raw data from the supplement using R's haven::read_dta(). There's no codebook so I'm not sure what all the columns represent, however.

I can also confirm that the enclosed .do files appear to contain code; however, I do not know STATA and cannot comment on the quality of the code.

Tables would benefit from more thorough captions. I often did not understand what the tables represented (particularly Table 5).

Experimental design

The author compares the performance of four tests or publication bias / p-hacking: Egger's test, TES, p-uniform, and the caliper test. I think that this is a worthwhile endeavor. Among these four tools, it would be useful to know which ones perform well under which conditions.

I was confused by the use of effect size Beta. I am familiar with Cohen's d, odds ratios, and Pearson's r. Isn't Beta equivalent to a Pearson r? It must not be because r cannot exceed zero. Please clarify what the effect metric is.

I was a little surprised by the sizes selected for N and K. Are these parameters intended to be representative of a particular research area? These Ns are larger than I would expect in my area of experimental psychology. Also, I think it is unusual for a substantive meta-analysis to have k = 100. Perhaps it is your intent to test an entire research literature for publication bias (e.g. 1000 studies from some journal) -- it is generally my opinion that these samples are too heterogeneous to be informative. Please justify the decision to use these Ns and Ks.

It is worth considering that the implemented form of p-hacking is only one form of p-hacking. P-hacking may come in many forms (selection among several dvs, selection among several treatment groups, optional stopping, outlier exclusion), all of which increase the Type I error rate to various degrees and may influence these tests in different ways. Thus, it is sensible to caution that the results may not generalize to all forms of p-hacking.

Validity of the findings

See general comments.

Additional comments

Unfortunately, I had a lot of trouble understanding the manuscript. Reorganization, use of subheadings, and pruning of unnecessary text would be very helpful.

1. Organization
The manuscript is very long and dense. It would be helpful to give it a fresh read and evaluate what is most important. Organization by subheadings, especially in the results section, would be very helpful.

Some technical details, such as how long the code took to run, are technically impressive but not of scientific importance. I'm also not sure that it's necessary to fully explicate the math behind each test when they are described in their respective publications.

Footnotes are also frequent and not all seem necessary. Some footnotes contain factual inaccuracies; for example, footnote 4 cites Nuijten et al. 2016 as an estimate of the prevalence of fraud, but this paper only checks the prevalence of p-values that don't match their test statistics (which can happen for benign reasons). I am also unsure about whether "publication bias is heavily punished" in any area except clinical trials.

2. Results
The results are confusing to me because they seem to describe many results in terms of unconditional probabilities, e.g. averaged across all conditions. This is confusing because many of these probabilities will depend heavily on the condition. For example, the frequency with which null results are censored will depend heavily on the true effect and the number of observations.

The regression model might help to address this concern, but I remained too confused about what data and conditions were in the model to learn anything from this section.

Additionally, I find the emphasis on dichotomous significance tests a little dissatisfying. A better question might be, how badly has the true effect been misestimated? You mention that "PU was only able to discover file-drawer behaviour under low underling true effects." (line 449). Perhaps at this N and this K, file-drawing does little to influence the results of a meta-analysis? Just how bad is the bias?

I don't understand Table 5.

3. Interpretation.
The Discussion and Conclusion seems very brief compared to the rest of the manuscript. I found this disappointing because there is a lot that is done in the Results section that was not clear to me. I was hoping that the Discussion would establish the meaning and importance of all these analyses and tables.

The author recommends the use of TES when publication bias selects for results of either sign. It is not clear to me that two-tailed significance selection is a hazard in the same way that one-tailed significance selection is. Consider that when publication selects for significant results of either sign, this doesn't always lead to bias in the meta-analytic effect size estimate. There is a loss of efficiency due to the censored studies in the middle, but the positive and negative results often cancel each other out. With this in mind, what is learned by a significant TES result? Does one learn that the naive meta-analytic result is necessarily wrong? Or is the idea that somebody has to be punished for file-drawering, even if the meta-analytic estimate is roughly unbiased? It doesn't seem very informative to scientific progress.

Some other corrections to the introduction may influence the interpretation of your results. Simonsohn and van Aert disagree (line 196) because they are considering different estimands and different meanings of heterogeneity. van Aert attempts to recover the overall mean of all studies whereas Simonsohn attempts to recover the true mean of only the significant studies. If it's Beta or Delta or Rho you intend to recapture, then van Aert is correct: p-uniform will overestimate the true mean under heterogeneity. You may need to reinterpret your p-uniform results accordingly.

Second, there are studies that have considered the FAT under heterogeneity. See https://www.ncbi.nlm.nih.gov/pubmed/12205693 and http://onlinelibrary.wiley.com/doi/10.1002/sim.1461/full.

I apologize that I must sound a bit slow given how much I did not quite understand. I assure you that I tried my best.

I always sign my reviews,
Joseph Hilgard

---

## Round 0.2 · Minor Revisions

· Academic Editor

Minor Revisions

I suggest you to take into consideration the comments of Joseph Hilgard. You could e.g. refer to future studies to address the (possible) limitation by choosing the p-value as evaluation criterion.

·

Basic reporting

No comment.

Experimental design

No comment.

Validity of the findings

No comment.

Additional comments

The revised manuscript successfully addresses my previous comments.

·

Basic reporting

It's fine.

I don’t mean to be picky, but I found myself struggling with the language at points. Line 316, commas are misplaced. If I understand it correctly, it should read as, “For β = 0.5 in the 50% publication bias condition only 35%; for β = 1, 15%; for β = 1.5, only 9%; and in the heterogeneous condition, 22% of the studies employed publication bias practices.”

Minor points.

Line 383: Don’t you mean compared to the file-drawering condition?

You may find it helpful to make explicit what is the predicted quantity in Table 3’s OLS regression. It was not clear to me. The number of analyses conducted? The mean p-value?

Graphs may also be useful – in general, I found it difficult to interpret these many tables and their many numbers.

Experimental design

I find myself wishing there was more attention to the bias in the actual effect size estimate. I think the attention on the p-value and the p-value alone is rather shortsighted. I understand that this suggestion would potentially lead to a rather different paper, however.

This unusual emphasis on the p-value alone leads to unusual statements such as line 306, “Because the heterogeneous effect condition does not allow an absolute bias measure, the p-value deflation factor was used for all decisions.” I find this strange – why not conduct random-effects meta-analysis and compare the meta-analytic mean (and tau) against the true mean (and tau)? Consideration of the p-value deflation factor alone seems odd to me. If there is some true effect, I don’t particularly care whether 5/10 studies find it or 10/10 studies find it so long as the resulting meta-analytic estimate and inference is correct.

Validity of the findings

You may also consider the nature of your file-drawer condition. This is a relatively mild form of publication bias in that each agent has only a (1-(.95^10)) = 40% chance of getting a significant result, and if they get no significant result, they publish a nonsignificant result. More commonly, simulations like this generate k studies and publish only some percentage of the nonsignificant results. Compared to social psychology, where almost every published result is significant, having 60% of published results be nonsignificant may be pretty modest bias.

Also, does it really make sense to compare the power of tests under file-drawering vs. p-hacking given that they inflict different amounts of bias in the meta-analytic result? I can see the utility of comparing these methods against each other, but to make inferences about the power of the method under one scenario, as compared to another scenario, seems inappropriate. To say that methods gained or lost power in going from p-hacking to file-drawering seems to overlook that the two practices inflict different amounts of bias.

The noted weakness of the caliper test is interesting, namely that under less-than-huge K, few studies fall within the caliper. But at Line 357, discussing high false-positive rates of caliper tests, can you provide evidence that the underlying true effect yielded an overrepresentation of just-significant values? That doesn’t sound like something that should happen, if I understand how p-values work. It is also curious that this happens at what appear to be specific levels of Beta, N, and K, rather than behaving in some predictable relationship.

I am a little wary of endorsement of the TES given the numerous criticisms of it that have been published —see https://alexanderetz.com/tag/test-of-excess-significance/ for a compendium.

Additional comments

The revised paper is rather easier to read, and the conclusions seem sensible. FAT works well under homogeneity and one-sided selection, but the CT and TES may work under heterogeneity and two-sided selection.

I have a litany of suggestions, which you are free to accept or to reject. I do not want to be the reviewer who asks for a completely different paper, but I think consideration of these points would make for a clearer, more informative paper.

---

## Round 0.3 · accepted · Accept

· Academic Editor

Accept

All comments of the external reviewers have been addressed diligently. I hope that your article will find broad adoption in the community.